# Increasing Measles Seroprevalence in a Sample of Pediatric and Adolescent Population of Tuscany (Italy): A Vaccination Campaign Success

**DOI:** 10.3390/vaccines8030512

**Published:** 2020-09-08

**Authors:** Beatrice Zanella, Sara Boccalini, Benedetta Bonito, Marco Del Riccio, Emilia Tiscione, Paolo Bonanni, Angela Bechini

**Affiliations:** 1Department of Health Sciences, University of Florence, 50134 Florence, Italy; beatrice.zanella@unifi.it (B.Z.); sara.boccalini@unifi.it (S.B.); benedetta.bonito@unifi.it (B.B.); emilia.tiscione@unifi.it (E.T.); paolo.bonanni@unifi.it (P.B.); 2Medical Specialization School of Hygiene and Preventive Medicine, University of Florence, 50134 Florence, Italy; marco.delriccio@unifi.it (M.D.R.); alessandra.ninci@unifi.it (W.G.D.); 3Meyer Children’s Hospital, 50139 Florence, Italy; antonino.sala@meyer.it; 4AUSL Toscana Centro, 50122 Florence, Italy; giovanna.mereu@uslcentro.toscana.it

**Keywords:** measles, elimination, vaccination, pediatric, adolescence, seroprevalence, Italy, vaccination coverage, vaccination effectiveness, MMRV vaccination

## Abstract

*Background*: Despite the National Plan for the Elimination of Measles and congenital Rubella (NPEMcR), in 2017, a measles outbreak occurred in Italy, due to sub-optimal vaccination coverage (<95%) for many years. Since that year, the anti-measles vaccination became compulsory in minors (0–16 years) for school attendance. The aim of our study was to assess the immunity/susceptibility against measles in a representative sample of pediatric and adolescent (1–18 years) residents of the province of Florence (Tuscany, Italy), and to compare these results with two previous surveys (2003 and 2005–2006). *Methods:* The enzyme-linked immunosorbent assay (ELISA) was applied for a qualitative measurement of anti-measles antibodies on 165 sera. The anamnestic and vaccination status was also collected. *Results*: No measles notification was reported. The overall seropositivity was 88.5%; mostly in the 5–9 years old subjects (97.9%). Among the 152 vaccinated, 92.1% were positive. The seropositivity persisted after many years since the last dose of vaccine and tended to be more long-lasting in those who had received two or three doses. The susceptibility towards measles decreased over time, reaching a lower value in the current survey (8.5%) than in 2003 (30.8%) and in 2005–2006 (25.5%). *Conclusions*: This study confirmed the anti-measles vaccination campaign success, which allowed for the increase in vaccination coverage and immunity levels against measles in the Florentine pediatric and adolescent population following the NPEMcR implementation.

## 1. Introduction

Measles is a highly contagious viral infectious disease known globally. Despite the availability of an effective and safe vaccine since 1960s, the mortality amongst young children (1–3 years old) remains high worldwide [1]. Although vaccination contributed to reduce the global measles deaths (79% in the period 2000–2018) and prevented 17.1 million measles-related deaths during 2000–2014, measles still occurs often in developing countries, especially in some parts of Asia and Africa [2].

The measles virus (MV) belongs to the genus Morbillivirus of the family *Paramyxoviridae*. It is an enveloped, single-stranded, negative-sense RNA virus [3]. Typically, MV is transmitted through saliva droplets over short distances but also by small aerosol particles that remain suspended for several hours in the air. The measles incubation phase is quite long: about 10 days before fever occurs (accompanied by cough, coryza, and conjunctivitis) and 14 days before rash appearance [4]. Measles is characterized by both seasonal epidemics and longer cycles. The first mainly occur in the winter because of an intensification of social risk factors (i.e., crowded enclosed places); the second, on the other hand, is mainly due to an accumulation of susceptible subjects who have never been vaccinated or previously contracted the disease [5]. The aim of the World Health Organization (WHO) was the elimination of this disease in five out of six regions within 2020. America is the only one to have reached this goal thus far [2]. Nevertheless, the USA experienced several outbreaks in 2018–2019, especially in the states of New York and New Jersey. Most cases were linked to unvaccinated travelers who brought back the virus from Israel, spreading the infection primarily in Orthodox Jewish communities [6].

Measles in still an endemic infectious disease in 14 European countries, including Italy. In the last 4 years, many European countries faced a measles epidemic outbreak. The European Centre for Disease prevention and Control (ECDC) is constantly monitoring the situation and, according to the last annual epidemiological report of 2019, 13,200 cases of measles were reported by 30 EU/EEA Member States. France, Romania, Italy, Poland and Bulgaria accounted for 65% of all reported cases in 2019, although their combined populations only represent approximately 37% of the EU/EEA population. Around 73% of the total cases of infected subjects were not vaccinated; this highlights the problem of vaccine hesitancy, which still represents one of the major public health concern [7,8]. The overall report rate was 25.4 cases per 1,000,000 population, which was lower than in 2018 and 2017 (34.4 and 35.5), but much higher than the rates observed in 2015–2016 (7.8–9) before the start of the epidemic in Europe. Furthermore, in 2019, there were 10 reported deaths due to measles (case–fatality: 0.09%) and the age-specific report rates decreased with increasing age, with unvaccinated children < 1 year and those aged 1–4 years most affected. Children below 5 years of age accounted for 28% of the cases, while adults aged 20 years and above accounted for 49% of the cases [2]. For the measles elimination goal to be reached, ECDC recommends many countries to increase coverage and uptake of their routine childhood immunization programs. They would also need to close immunity gaps in adolescents. In Italy, a surveillance system to efficiently monitor the measles spread has been established in 2014. In 2017, due to low levels of vaccination coverage (VC), a measles outbreak occurred in Italy, mirroring the European scenario [9,10]. Almost 5000 cases (4991 cases) have been reported (mostly < 1 year old and young adults >15 years old), either unvaccinated or vaccinated with a single dose [11]. According to the last report published in January 2020, 1627 cases were reported in 2019 (incidence is 27 cases/million). In the same year, all Italian regions have reported cases; however, 58% of them were reported to be in the Lazio and Lombardia regions. The median age was 27 years old, and 62 cases in children of <1 year old have also been reported. The report states that 86.2% of the cases were not vaccinated at the time of infection, and 31% developed at least one complication. Furthermore, 96 cases amongst the healthcare workers and 43 amongst school system workers have been reported [12,13]. The average measles immunization coverage at the 24th month in Italy is 93.2%; importantly, Tuscany is the Italian region with the highest immunization coverage at 24 months [14]. The measles epidemiology in Tuscany follows the Italian overall trend. Measles incidence dropped from 383 cases in 2017 to 90 cases in 2018, and the median age was similar to the national one (29 years vs. 27 years), which considerably increased compared to 1994 (about 16 years old) [15,16]. The lowest VC at 24 months in Tuscany was reached in 2001 (83.6%), peaking then in 2019 (95.3%) due to the introduction of the measles vaccine as mandatory for the school attendance in 2017 [15,17]. The measles outbreak in Italy, and specifically in Tuscany, has been monitored over the years, highlighting how the vaccination promotion programs have greatly helped in increasing the immunization coverage [18,19,20]. The most effective tool to prevent measles is through vaccination, which provides a long-lasting immunity for decades, even without boosting after the primary immunization course [21].

The current available vaccines in Italy are attenuated live vaccines in two different formulations: trivalent MMR (measles–mumps–rubella) and quadrivalent MMRV (measles–mumps–rubella–varicella). In Italy, the vaccination schedule recommends one dose between the 13° e 15° month, and the second at 5–6 years old. For unvaccinated adolescents and susceptible adults, two doses at 4 weeks distance are recommended [22]. Therefore, there is a strong need to implement the vaccination program and ensure a VC > 95%, recommended by WHO, in order to protect people who cannot be vaccinated via herd immunity.

The aim of our study was to assess the immunity/susceptibility to measles in a representative sample of a pediatric and adolescent (1–18 years) resident population in the province of Florence. Sera collected at the blood sampling center of the Meyer Children’s Hospital in Florence have been analyzed aiming to find anti-measles IgG antibodies. This study is part of a wider project which include the investigation of other infectious diseases like rubella, varicella, hepatitis A and B and tetanus [23]. Despite the compulsory vaccination order since July 2017, epidemics are still observed. Additionally, another purpose of this study was to compare the obtained results with the data of the previous seroepidemiological surveys carried out in 2003 and 2005–2006 within the same geographical area in order to describe the temporal trend of the immunity level against measles for the pediatric and adolescent population. Furthermore, we wanted to assess if the presence of the antibodies in the sera was due to a natural infection or due to the vaccination, and to assess the VC with the administration of one or two vaccine doses.

## 2. Materials and Methods

The recruitment and sera collection methods adopted in this study were the same as those used by Zanella et al., 2020, in which hepatitis B seroprevalence was investigated [23]. The collection of blood samples took place during the period December 2017–April 2018 at the blood sampling center of Meyer Children’s Hospital. All parents or guardians of the enrolled subjects gave written consent for the inclusion before enrollment. The study was conducted in accordance with the Declaration of Helsinki, and the protocol was approved by local ethics committees (Project identification code: DSS-UNIFI, n. registro pareri 98/2017). In a post hoc analysis, the calculated sample size was 162 sera, estimating the expected seroprevalence of anti-measles virus equal to 85%, an accuracy of 5.5% and a confidence level of 95%. This value represents 0.1% of the resident population of Florence aged 1–18 years among 166,644 subjects in 2017 in the same age group [24]. This is also proportionally related to the population composition for both age group and gender. Thus, no further standardization was required.

The National Registry of Notifications for Infectious Diseases (SIMI software: Epi Info, Rome, Italy) was consulted to assess the anamnestic status of each subject. Through the vaccination registers, SISPC (Collective Prevention Healthcare Information System; Consortium Metis, Tuscany, Italy) and Caribel (Aster, Tuscany, Italy) (the current and the previously used VC software in the Tuscany region, respectively), we retrieved the vaccination status for measles, the number of measles vaccine doses, the year of the last dose, and, when available, the type of the last administered vaccine for each enrolled subject. The serological profile towards measles obtained in the present study was compared with two previous serological surveys carried out in 2003 and 2005–2006 within the area of Florence [19].

We excluded non-residents in the province of Florence, immunocompromised patients, subjects under immunosuppressive treatment, those with an acute infectious disease (rubella, varicella, hepatitis A, and hepatitis B) in the previous two weeks, and those who had received a blood transfusion within the six months prior to the study. All the collected blood samples were centrifuged (1600 rpm at 4 °C), and the recovered sera were stored at −20 °C until tested for measles.

All the sera were tested for anti-measles antibodies.

The commercial Enzygnost^®^ Measles Anti-virus/IgG enzyme-linked immunosorbent assay (ELISA) (Siemens Healthcare Diagnostics Products GmbH–Germany) was used to perform a qualitative measurement of anti-measles antibodies. The cut-off values are as below:Anti-measles/IgG negative ΔA < 0.100 (cut-off);Anti-measles/IgG positive ΔA > 0.200;Anti-measles/IgG equivocal 0.100 ≤ ΔA ≤ 0.200.

Where ΔA is the OD value obtained for each sample—the OD value of the same sample containing the control measles antigen (OD: optical density).

Equivocal samples were tested a second time. If the result came out as positive, the sample was considered positive; on the other hand, if the result was negative, the sample was considered negative. If the result was still equivocal, the final outcome of that sample was equivocal.

For the analysis, subjects were divided by age group, sex, and nationality. Particularly, children who were born in Italy (yet not holding the Italian citizenship) were considered as Italians; the remaining were foreigner or holding a dual-nationality (not Italian).

The statistical analysis on the seroprevalence was performed assuming as non-positive the subjects found to be negative or equivocal. Differences between nationality, sex, age, vaccination status, number of doses of vaccine and time elapsed since the last dose of vaccine were evaluated. Furthermore, statistical significances were assessed with the two-tailed chi-squared test or Fisher’s exact test when the chi-squared test was not applicable. Simple logistic regression analysis were performed for the vaccinated population considering serological status (positive or non-positive) as the outcome (dependent) variable to assess if the time elapsed since the last dose of vaccine, the age group and the number of doses of vaccine were significantly associated as independent variables. For all the analyses, a *p*-value < 0.05 was considered significant. The statistical analyses were conducted using RStudio 1.2.5033 (RStudio Team, 2019. RStudio: Integrated Development for R. RStudio, Inc., Boston, MA; URL: http://www.rstudio.com).

## 3. Results

The present study included 165 subjects aged 1–18 years divided into four age groups, as shown in Table 1. Males and females represented the 53.3% and 46.7% of the study population, respectively. Most of the enrolled subjects were Italian citizens (90.3%), the remaining part consisted of subjects having not-Italian citizenship: either dual nationality or foreigner (9.7%). The participants resided in 35 different districts of Florence, and about 48% of them were living in the city of Florence.

### 3.1. Measles Antibodies Qualitative Measurement

Table 2 shows the percentage of the positive, negative and equivocal subjects in relations to their sex (male and female) and their nationality (Italian or not Italian). Amongst the 165 samples, 146 were positive (88.5%), 14 were negative (8.5%) and 5 were equivocal (3%). There was no significant difference in the percentage of seropositive subjects (*p* > 0.05) among the male and female groups, and among the Italian and not-Italian groups.

The majority of the positive subjects were in the age group 5–9 years old (47/48 subjects corresponding to 97.9%) followed by the 1–4 years group (35/40 subjects corresponding to 87.5%), 15–18 years group (23/27 subjects corresponding to 85.2%) and finally the 10–14 years group (41/50 subjects corresponding to 82%) (Figure 1). Logistic regression analysis was used to evaluate the odds of being positive vs. not positive in different age groups, resulting in no significant difference in the percentage of seropositive subjects among the age groups (*p* > 0.05).

Figure 2 describes the distribution (%) of the age groups in the positive, negative and equivocal sera. In the positive ones, the most represented age group is the 5–9 years old group (32.2%), followed by the 10–14 years old group (28.1%), the 1–4 years old group (24%) and the 15–18 years old group (15.8%). Amongst the negative results instead, the two age groups most represented are the 1–4 years old and the 10–14 years old groups, both 35.7%, followed by the 15–18 years old group (21.4%) and the 5–9 years old group (7.1%). The smallest number of equivocal sera (5) is in the two oldest age groups.

### 3.2. Measles Notification, Vaccination Status and Seroprevalence Assessment

According to SIMI, none of the enrolled subjects were notified for measles.

The secondary aim of this study was to assess the vaccination status of each subject and relate it to the seroprevalence found (Table 3). Most of the enrolled population was vaccinated (92.1%), and the vaccine administered was MMR and/or MMRV. Among the 152 vaccinated subjects, 140 were positive (92.1%), while 8 were negative (5.2%) and 4 were equivocal (2.7%). The most pronounced seroprevalence in the vaccinated was in the 5–9 years old group (100% of positive samples), followed by the 1–4 (97.1%), 15–18 (89.2%) and 10–14 years old groups (84.1%).

On the other hand, the unvaccinated subjects were 13/165 (7.9%). Among them, six were positive (46.1%), six were negative (46.1%) and one was equivocal (7.8%). The non-positivity status (negative and equivocal) was represented by four subjects aged 1–4 years, one subject aged 5–9 years and two subjects aged 10–14 years. As expected, the vaccination status was found to be strongly associated with the seroprevalence status (*p* < 0.0001).

Furthermore, we retrieved the number of vaccine doses the subjects had received in their lifetime, and we evaluated the seroprevalence trend. Figure 3 shows that 98% of the subjects who had received one dose were positive, and 2% were negative to serological test: 49/50 and 1/50, respectively. Among subjects who had received two or three doses of vaccine, 89.2% (91/102) resulted positive, 6.9% (7/102) negative and 3.9% (4/102) equivocal. No significant difference in the seroprevalence status related to the number of received doses was found (*p* >0.05).

Figure 4a–c show the number of positive, negative and equivocal subjects in relation to the distance from the last dose of vaccine received. Figure 4a shows the overall scenario, in which measles seropositivity remains high up to eight years since the last dose in subjects who had received both the doses. In Figure 4b, we can observe that all those vaccinated with just the first dose are positive (100%) except one, who received the vaccination in the same year of recruitment. On the other hand, although subjects who received two/three doses present a slightly higher number of negative/equivocal subjects, the seroprevalence appears to be long-lasting up to 13 years since the last dose (Figure 4c). The negative and equivocal subjects are among those who received two/three doses, from 6 to 14 years before the serological test (Figure 4c).

## 4. Discussion

This study aimed to assess the anti-measles seroprevalence in a pediatric and adolescent (1–18 years) population resident in the province of Florence, enrolled between 2017 and 2018. Regarding the nationality, our sample population reflects the overall Italian situation: almost 17% is represented by children with a non-Italian citizenship, where in Italy children born from at least one foreign parent were ~20% in 2018 [25]. There was no notification of measles disease in our enrolled sample; this agrees with the fact that in the recent Italian epidemics, the average age was 27 years old [12].

Considering the seroprevalence results, 88.5% of the total number of subjects presented anti-measles antibodies, whilst 8.5% did not. A previous study focusing on the measles seroprevalence in Tuscany [19] reported that susceptible people (1–19 years old) were about 30.8% in 2003 and about 25.5% in the period of 2005–2006. It is notable how the susceptibility trend decreases over time reaching the lowest point in the current survey (8.5%). In particular, the seroprevalence trend considerably increased in the age groups 1–4 (87.5%) and 5–9 years old (97.9%) compared to the previous surveys performed in 2005–2006 (respectively 76.2% and 84.2%) and in 2003 (62.1% for the 1–4 years old group, and 60% for the 5–9 years old group). The recent decrease in the susceptibility towards measles in the pediatric and adolescent population of Tuscany, together with the increased measles seroprevalence, is mainly attributable to all the preventive activities performed in Italy following the adoption of the “National Plan for the Elimination of Measles and congenital Rubella (NPEMcR)” [20]. The first national plan was realized in all Italian regions in 2003, and it had been continuously implemented up to the last NPEMcR 2010–2015, approved in 2011 [22]. Even if measles outbreaks continued to occur, our results confirm the great impact of the immunization activities in the pediatric and adolescent population in Tuscany, where the WHO thresholds for the elimination of measles in the European Region have been achieved for the first time in the three age groups. According to our results, the percentage of susceptible subjects aged 1–4 years old (12.5%) and 5–9 years old (2.1%) is under the threshold established for the European region to reach the elimination of measles, which is fixed at 15% for children aged 1–4 years and 10% for those aged 5–9 years. [26].

In Tuscany the lowest VC at 24 months was reported in 2001 (83.6%), increasing progressively until 2009 (92.6%), then decreasing again, reaching 88.7% in 2015. The reasons behind the missed measles vaccination in 2017 in Italy were reported to be “total dissent” (4.8%) and “found/contacted but did not attend the appointment” (2.3%); moreover Sicily, Friuli Venezia-Giulia, Veneto, Marche and the Autonomous Province of Trento presented a higher percentage of missed measles vaccination compared to the national value [8]. The missed vaccination issue is part of the wider phenomenon of the parental vaccine hesitancy (“no-vax movements”) in which religious and moral beliefs, complacency and skepticism made the perception of vaccine as scary and unnecessary [7,27]. In order to face this negative sentiment and to contrast the decreasing trend in immunization coverage in 2017, an Italian national law, the “Vaccine Decree”, [22] introduced measles vaccination as mandatory to enter the school for subjects between 0 and 16 years of age. The lowest percentage of positive subjects was found in the age group 10–14 (82%) followed by the 11–18 years old group (85.2%). This result was expected since the lower antibody presence in these groups compared to the previous two may be due to the longer distance since the last vaccine dose, as demonstrated by the measurement of measles-neutralizing antibodies in the subjects’ sera at different time points after vaccination [28,29]. It could also reflect the immunization coverage data of these birth cohorts.

Taking into account the vaccination status of the subjects, it is notable that most of them had received at least one dose of measles vaccine (98%). However, not all of them were positive to the anti-measles ELISA test; thus, 8/152 were negative despite vaccination. This could be explained by the fact that not enough time had gone by to develop anti-measles IgG antibodies. As a matter of fact, IgG antibodies typically appear 12–15 days after vaccination and peak at 21–28 days. Other reasons could be found in the presence of inhibitory maternal antibodies or in the immunological maturity of the toddler (case of the one-year old boy in our sample). Moreover, as known, the efficacy of the first measles dose is 95%; in this condition, 5% of vaccinated subjects receiving only one dose remained unprotected. Thus, this is why a second measles vaccine dose is needed to catch up non-responders to the first dose of vaccine. Lack of measles antibodies despite vaccinations could also depend on the vaccine dose and strain of the virus contained in the vaccine [21,30]. Importantly, it has been widely demonstrated that those vaccinated lose the antibody titer over time [29,31]; thus, it was recently reported that even people who received two doses of vaccine developed measles in California [32] and Spain [33]. Among the 13 unvaccinated subjects, 6 of them were found to be positive according to the ELISA test. Notably, none of them were reported to have had the disease. Probably, either the healthcare worker did not correctly record the vaccination, or the disease was underreported or misreported. The level of measles notification varies greatly not only between Italian regions but also by the age of the subject, leading to misinterpretation of the epidemiological data [34,35]. Four of these subjects have a non-Italian nationality, coming from the Philippines, Senegal and Burkina Faso. For the latter two countries, data on VC are high (85–90%), and measles report rates are low [36,37]; whereas in the Philippines, the VC dropped in the last years, reaching values below 70%, and measles cases were highly reported, especially in the years 2014 and 2018 [38]. Thus, possibly the Philippine child developed measles disease in his own country or missed the vaccine dose; however, children from Senegal and Burkina Faso probably had been administered the vaccination or had the disease and, these were not appropriately reported. Moreover, 4/152 subjects who had received two vaccine doses resulted in being weakly protected according to the equivocal outcome of the serological test. It is likely that they developed either a weak response to the vaccine or they showed a low antibody detection due to the distance since the last dose (six to twelve years) [28].

Analyzing the administered doses in the four age groups, we can observe that the whole 1–4 year old group was vaccinated. In the 5–9 years old group, 67% had received two doses, and one-third had received one. These results are consistent with the Italian vaccination schedule, which recommends a second dose between 5–6 years old before school entry. Only one subject in the 10–14 years old group had received just one vaccine dose.

The highest percentage of anti-measles seroprevalence was found in children who received one dose of vaccine (98%) and, more recently, was followed by those received two/three doses (89.2%), even after a long time (up to 13 years) prior to the serological test, as already observed in the literature [39,40]. Despite the fact that the percentage of seropositivity in subjects who received two or three doses was lower than the percentage measured in those who received just one dose, this result was not confirmed by the statistical analysis carried out, and it is reasonable to consider it a vaccination campaign success. Overall, the number of positive subjects remains high until the seventh–eighth year since the last dose. Notably, the seroprevalence is very high in almost all the subjects vaccinated with the first dose; moreover, despite the presence of negative and equivocal subjects vaccinated with two or three doses, the presence of measles antibodies lasts up to 13 years from the last dose, confirming the long-lasting effectiveness of the vaccine.

## 5. Conclusions

Our study assessed the measles seroprevalence in a sample of sera representative of the pediatric and adolescent population residing in the province of Florence. The overall seropositivity was 88.5%, and highest immunity level were found in the 5–9-year old subjects (97.9%). Furthermore, comparing these results with two previous serosurveys carried out in 2003 and 2005–2006, the current study highlighted a dramatic decrease in susceptibility towards measles (8.5%), with a lower value than in 2003 (30.8%) and in 2005–2006 (25.5%).

Vaccinating against measles is a cost-effective preventive intervention, and it is recognized by WHO as a public health priority in order to achieve the goal of its elimination. Importantly, in the recent COVID-19 pandemic, a hypothesis which is still under assessment has arisen regarding the possibility that vaccination against measles may work as a preventive tool for coronavirus as well. Studies speculate that the lower incidence and mortality rates of the SARS-CoV-2 infections among children below 9 years old could derive from a cross-protective effect by the antibodies developed after the anti-measles vaccination. [41]. Whilst currently there are not sufficient information about it, if research studies confirm these preliminary data, promotion of the anti-measles vaccination could potentially be used as a further preventive strategy while a safe and effective vaccine against SARS-CoV-2 is still pending.

This study confirms that the promotion of measles vaccination campaigns were successful when considering the Florentine pediatric and adolescent population. The effort in increasing the vaccination coverage resulted in Tuscany being the leader region in Italy with the highest measles immunization coverage at 24 months [39]. In addition, serological studies allow the monitoring of immunization levels in a population over time. Our results highlighted a marked increase in the immunity levels against measles among the pediatric and adolescent population residing in the province of Florence when compared with two previous serological surveys carried out in the same area. The low susceptibility rates measured in the first two age groups may represent the step towards the elimination of measles at local and regional levels. Finally, in order to reach the complete nationwide measles elimination, it is essential to maintain a high level of immunization against measles. This can be done by not only reinforcing vaccination strategies, but also increasing the quality of surveillance. Seroprevalence studies are confirmed as an essential tool to monitor the immunization status of the general population at regional and national levels.

## Figures and Tables

**Figure 1 vaccines-08-00512-f001:**
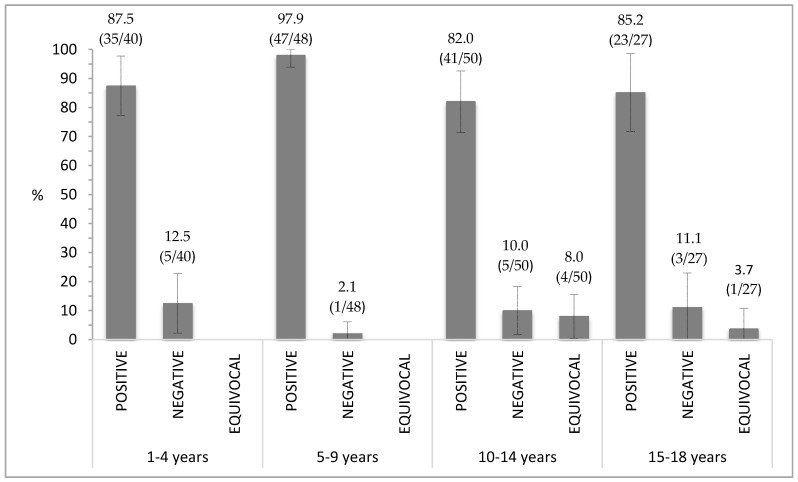
Percentage of measles immunity distribution in the age groups.

**Figure 2 vaccines-08-00512-f002:**
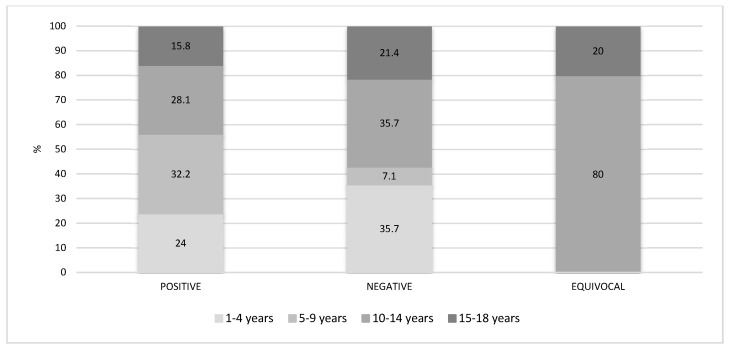
Age group percentage distribution of the positive, negative and equivocal subjects.

**Figure 3 vaccines-08-00512-f003:**
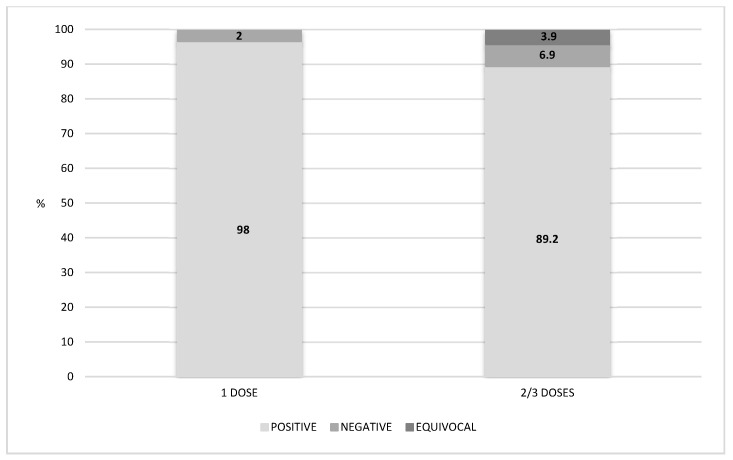
Percentage distribution of positive, negative and equivocal subjects according to the number of vaccine doses received.

**Figure 4 vaccines-08-00512-f004:**
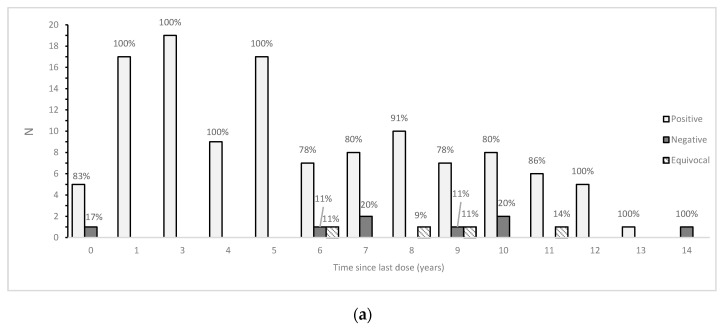
Seroprevalence in relation to time since last dose received. Total doses (**a**), one dose (**b**), two or three doses (**c**). Percentages are shown above each histogram. Logistic regression analysis was performed in the vaccinated population considering the serological status as the outcome (dependent variable) and the time elapsed since the last dose of vaccine as the independent variable. Time elapsed since last dose resulted in being associated with the seroprevalence status (OR 1.28; 95% CI: 1.08–1.55; *p* < 0.01).

**Table 1 vaccines-08-00512-t001:** Enrolled subjects divided into age groups.

Age Group (Years)	Enrolled Subjects (*N*)
1–4	40
5–9	48
10–14	50
15–18	27
Total	165

**Table 2 vaccines-08-00512-t002:** Anti-measles seroprevalence in the enrolled population. (*n*/*N*: number of subjects/total).

Anti-Measles Seroprevalence			
Group	POSITIVE % (*n*/*N*)	NEGATIVE % (*n*/*N*)	EQUIVOCAL % (*n*/*N*)
Overall	88.5 (146/165)	8.5 (14/165)	3 (5/165)
Male	90.8 (79/87)	6.9 (6/87)	2.3 (2/87)
Female	85.9 (67/78)	10.3 (8/78)	3.8 (3/78)
Italian	90 (134/149)	7.3 (11/149)	2.7 (4/149)
Not-Italian	75 (12/16)	18.8 (3/16)	6.2 (1/16)

**Table 3 vaccines-08-00512-t003:** Status of the enrolled population.

Vaccination Status	Age Group (Years)	Positive % (*n*/*N*)	Negative % (*n*/*N*)	Equivocal % (*n*/*N*)	Total % (*n*/*N*)
Vaccinated		92.1 (140/152)	5.2 (8/152)	2.7 (4/152)	92.1 (152/165)
	1–4	97.1 (34/35)	2.9 (1/35)	0 (0/35)	23 (35/152)
	5–9	100 (47/47)	0 (0/47)	0 (0/47)	30.9 (47/152)
	10–14	84.1 (37/44)	9.1 (4/44)	6.8 (3/44)	29 (44/152)
	15–18	89.2 (33/37)	8.1 (3/37)	2.7 (1/37)	17.1 (26/152)
Unvaccinated		46.1 (6/13)	46.1 (6/13)	7.8 (1/13)	7.9 (13/165)
	1–4	20 (1/5)	80 (4/5)	0 (0/1)	38.4 (5/13)
	5–9	0 (0/1)	100 (1/1)	0 (0/1)	7.8 (1/13)
	10–14	66.6 (4/6)	16.7 (1/6)	16.7 (1/6)	46 (6/13)
	15–18	100 (1/1)	0 (0/1)	0 (0/1)	7.8 (1/13)

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
