# Peer review of "Increasing Measles Seroprevalence in a Sample of Pediatric and Adolescent Population of Tuscany (Italy): A Vaccination Campaign Success"

_vaccines, 2020, doi:10.3390/vaccines8030512_

Round 1

Reviewer 1 Report

I read the manuscript by Zanella and co-workers.

I apologize, but the manuscript suffers from notable shortcomings, including mainly the limited number of the sample (152 vaccinated and 13 unvaccinated) which precludes a serious differentiation in age classes and according to gender. This does not allow an adequate statistical analysis. Further, the results do not lead to an increase in knowledge in the field of measles vaccination.

It is widely demonstrated that the time elapsed between the antibody analysis and the last measles vaccination is decisive in finding positive antibodies or not.

Other minor

line 80: all unvaccinated.. It is not really true

line 81-83: It is not clear which years refers the sentence

Figure 4: The absolute number does not seem indicative to me (also given the variable numerosity); the percentage would be more informative.

Reviewer 2 Report

This manuscript reports on a population seroprevalence study using ELISA for measles virus in the Tuscany region of Italy.

While the manuscript is understandable, editing for English grammar and style would improve the readability.

Line 36: should change 2/3 to 2 or 3

The introduction would be improved with a brief introduction to the relationship between elapsed time since MV vaccination and immunity. This is a well-studied topic and is heavily discussed later in the paper.

Line 57-58: awkward, please re-write this sentence for clarity

Line 60: It might be worth mentioning that a 2018-2019 MV outbreak in the US nearly jeopardized the WHO elimination standing of the America region.

Line 78: Needs a citation for attributing the outbreak to low levels of VC, or alternatively rephrase this as the speculative cause.

Methods paragraph lines 133-138: How many samples were excluded?

Line 147: How many samples (what percent) had an “equivocal” measurement during the first test?

Figure 1 would be improved by the inclusion of error bars to indicate 95% confidence intervals.

It would also be nice if the number of participants (n) were used on the bars in figure 1, 2, and 3 instead of the percentage which can be determined directly from the y-axis of the graphs. The inverse is true for figure 4.

Line 219: Sentence fragment, please fix to make the statement understandable.

Line 221: Please expand on the three-dose vaccine schedule. In the introduction (line 97-99), a two-dose schedule is discussed.

Line 229-230: This statement does not follow from the data presented. In Figure 4B, the number of seropositive individuals decreases based on the population sampling used, but the percentage of seropositive individuals remains 100% for all time points presented other than 0 years. The fact that one dose seems to retain a higher level of seropositivity at years 6-8, compared 2/3 doses, is discussed in the paragraph from lines 318-326, however the same unsupported conclusion is presented in the sentence on lines 325-326. It is a bit misleading that figure 4 changes from overlapping bars with a y-axis displaying percent in the previous figures to single bars with number of individuals. Presenting the data this way  highlights more of a trend to have sampled cases with less years since last dose, which is confusing since the discussion of the figure focuses on a trend towards decreased seropositivity with elapsed time since vaccination.

Line 314: using the fraction 2/3 when discussing number of doses adds confusion. Please consider replacing this with 67% to avoid confusion with 2/3 being used as an abbreviation for “2 or 3 doses”.

Line 329-336: This is an untested hypothesis. Although there is an ongoing clinical trial to analyze a potential link between MV and SARS2-CoV, the relationship is over-stated here. Please re-write this section to make it clearer that there is not sufficient data currently to support this claim.

Reviewer 3 Report

Listed below are a few specific editing changes but a more thorough reading for English is required!

Line 79….. started in --change to occurred in

Line 79….. A number of 4491 cases ----change to 4491 cases

Line 84…The 86.2% of the cases was not vaccinated ---change to The report states that 86.2% of the cases were not vaccinated Line 85-86 …. Sentence need rewording!

Line 220-221… dose was -change to dose were Line 241…elapsed since last dose to be associated.. needs editing?? Line 309-311….  Needs rewording!!

Line 346…. essential to maintain high the level----change to essential to maintain a high level

Suggestion: Conclusion section---Summarize the data presented in the paper followed by the statement regarding COVID-19.

Round 2

Reviewer 1 Report

I have not further comments.